# Long-Term Exposure to Fine Particulate Matter and the Risk of Chronic Liver Diseases: A Meta-Analysis of Observational Studies

**DOI:** 10.3390/ijerph191610305

**Published:** 2022-08-18

**Authors:** Jing Sui, Hui Xia, Qun Zhao, Guiju Sun, Yinyin Cai

**Affiliations:** 1Research Institute for Environment and Health, School of Emergency Management, Nanjing University of Information Science and Technology, Nanjing 210044, China; 2Key Laboratory of Environmental Medicine Engineering, Ministry of Education, School of Public Health, Southeast University, Nanjing 210009, China; 3Institute of Atmospheric Environmental Economics, Nanjing University of Information Science and Technology, Nanjing 210044, China

**Keywords:** fine particulate matter, chronic liver disease, meta-analysis

## Abstract

Although fine particulate matter (PM2.5) is a known carcinogen, evidence of the association between PM2.5 and chronic liver disease is controversial. In the present meta-analysis study, we reviewed epidemiological studies to strengthen evidence for the association between PM2.5 and chronic liver disease. We searched three online databases from 1990 up to 2022. The random-effect model was applied for detection of overall risk estimates. Sixteen eligible studies, including one cross-sectional study, one retrospective cohort study, and 14 prospective cohort studies, fulfilled inclusion criteria with more than 330 thousand participants from 13 countries. Overall risk estimates of chronic liver disease for 10 μg/m^3^ increase in PM2.5 was 1.27 (95% confidence interval (CI): 1.19–1.35, *p* < 0.001). We further analyzed the relationship between PM2.5 exposure and different chronic liver diseases. The results showed that increments in PM2.5 exposure significantly increased the risk of liver cancer, liver cirrhosis, and fatty liver disease (hazard ratio (HR) = 1.23, 95% CI: 1.14–1.33; HR = 1.17, 95% CI: 1.06–1.29; HR = 1.51, 95% CI: 1.09–2.08, respectively). Our meta-analysis indicated long-term exposure to PM2.5 was associated with increased risk of chronic liver disease. Moreover, future researches should be focused on investigating subtypes of chronic liver diseases and specific components of PM2.5.

## 1. Introduction

Fine particulate matter (PM2.5), responsible for most air pollution, has been increasingly affecting human health with exploding urbanization [1]. PM2.5 is defined as an ambient particulate matter with an aerodynamic equivalent diameter of ≤2.5 µm, mainly including organic matter, carbon, various metal compounds, nitrates, and sulfates [2]. Moreover, PM2.5 is with small particle size and large surface area, which results in absorbing toxins easily [3]. PM2.5 enters the circulation and travels to several organs after infiltrating the alveoli of lungs [4]. The International Cancer Research Center (IARC) has classified inhalable PM2.5 as the first class of carcinogen [5].

Chronic liver disease has become one of the leading causes of death worldwide with the increasing number of cases [6]. Chronic liver disease consists of various liver diseases, such as nonalcoholic fatty liver disease (NAFLD), alcoholic liver disease, liver cancer, cirrhosis and so on [7]. In vivo and in vitro studies have reported that inflammation, oxidative stress, gut microbiota dysbiosis and insulin resistance may be underlying pathogenesis of chronic liver disease [8,9,10,11].

An epidemiological study by Copeland et al. demonstrated that inflammation was identified as a signature of liver fibrosis and elevated risk for liver disease progression [12]. In a randomized study, antioxidant supplementation improved insulin sensitivity and reduced anthropometric parameters of NAFLD patients [13]. PM2.5 may affect the occurrence and progression of chronic liver disease through the mechanisms outlined above.

Current meta-analyses of PM2.5 and chronic diseases primarily focus on the morbidity and mortality of respiratory system diseases, cancer, cardiovascular and cerebrovascular disease [14,15,16,17,18]. In the literature, there are four meta-analyses [19,20,21,22] which have examined the association between PM2.5 and the incidence or mortality of liver cancer in nine epidemiological studies. In the above meta-analysis studies, only Wu et al. reported that PM2.5 was significantly associated with the incidence and mortality of liver cancer (hazard ratio (HR) = 1.28, 95% confidence interval (CI): 1.15–1.41; HR = 1.21, 95% CI: 1.13–1.29, respectively) [22], while others only examined the mortality of liver cancer [19,20,21]. The association between chronic liver disease and PM2.5, other than liver cancer, has been detected in epidemiological studies. Recently, a cross-sectional study showed that a 10 μg/m^3^ increase in PM2.5 increased metabolic dysfunction-associated fatty liver disease with odds ratio (OR) (95% CI) of 1.29 (1.25–1.34) [23]. Another prospective cohort study with 58,026 participants over 22 years presented a 1 μg/m^3^ increase in PM2.5 levels increased NAFLD with HR (95% CI) of 1.06 (1.04–1.07) [24]. Therefore, our present meta-analysis aims to investigate the association between PM2.5 and the risk of chronic liver disease based on observational epidemiological studies and various subgroup analyses to strengthen results based on new evidence.

## 2. Materials and Methods

### 2.1. Data Sources and Searches

We searched PubMed, Web of science, and the Cochran Library in May 2022 using common keywords related to PM2.5 and incidence and mortality of chronic liver diseases without language restriction. Searches were limited to original research articles published in between January 1990 and May 2022. The keywords were: “particulate matter “, “fine particulate matter”, “air pollution”, “atmospheric particulate matter” “ambient fine particle”, “ambient PM2.5”, and “PM2.5” for exposure factors, AND “Liver Disease”, “Liver Dysfunction”, “Nonalcoholic Fatty Liver Disease”, “NAFLD”, “Nonalcoholic Fatty Liver”, “Fatty Liver”, “Cirrhosis”, “Liver cirrhosis”, “Liver Fibrosis”, “Nonalcoholic Steatohepatitis”, “Liver Neoplasm”, “Hepatic Neoplasm”, “Liver Cancer”, “Hepatocellular Cancer”, “Hepatic Cancer”, “HCC”, and “Liver Cancer” for outcome factors. We also reviewed the reference lists of the original documents to identify additional pertinent data.

### 2.2. Study Selection and Eligibility

The selected inclusion criteria were as follows: (1) observational epidemiological studies, (2) investigation of the association between PM2.5 and the incidence and mortality of chronic liver diseases, (3) data of PM2.5 exposure levels were collected from monitoring station, horizontal–vertical locations, or satellite, and (4) outcome measures with adjusted relative risk (RR) or hazard ratio (HR) and 95% CI reported. We included comprehensive analysis to re-calculate duplicated or shared data appeared in two or more analyses [19,20]. The exclusion criteria were as follows: (1) reviews, letters, and reports, (2) animal- or cell-related data, (3) studies without subdivision of the types of particulate matter, (4) studies without PM2.5 increment data, (5) data collection time less than one year, and (6) studies on acute liver disease but not chronic liver disease [19,20,21,22]. Based on the inclusion criteria, J.S. and H.X. independently assessed the eligibility of studies. Any disagreements between evaluators were solved by mutual decision.

### 2.3. Data Extraction

A further collection from eligible articles were performed by J.S. and Q.Z. The collected data in each article were as follows: the publication year, first author name, type of study, study period, location of the study, number of participants, type of liver disease, adjustment variables, and adjusted RR/HR, 95% CI, etc.

### 2.4. Literature Quality Assessment

We estimated the quality of eligible studies with Newcastle–Ottawa Scale (NOS) [25]. We defined low-quality or high-quality studies according to the average score. The literature with a score higher than the average score was defined as high-quality.

### 2.5. Statistical Analyses 

We standardized the PM2.5 increment to a 10 µg/m^3^ increment, and recalculated the RR/HR with the following Formula [26]:$${\mathrm{RR}}_{\mathrm{Standardized}}=e^{\left(\frac{\mathrm{Ln}\left({\mathrm{RR}}_{\mathrm{origin}}\right)}{{\mathrm{Increment}}_{\mathrm{origin}}}\times {\mathrm{Increment}}_{\mathrm{Standardized}}\right)}$$
where RR is the relative risk, and Ln is the log to base e. Cochran’s Q and I^2^ were performed to evaluate heterogeneity in selected studies [27]. If *p* value for heterogeneity was less than 0.10 or I^2^ value was <50%, fixed effects model (FEM) was applied for the null hypothesis. Otherwise, we applied the random effects model (REM). Begg’s funnel plot and Egger’s test were used to calculate publication bias [28]. If the funnel plot was asymmetry or the *p* value for Egger’s test was less than 0.05, it indicates that there is publication bias in enrolled studies. We used Stata statistical software 11.0 to conduct all statistical analyses. 

## 3. Results

### 3.1. Eligible Studies

A flow diagram of literature screening process is shown in Figure 1. We identified a total of 7938 studies with literature retrieval strategies. After excluding 252 duplicate articles, 3801 articles were screened preliminarily. After reviewing titles and abstracts of above articles, 3699 articles were considered ineligible and excluded according to the inclusion and exclusion criteria. Finally, 102 articles were reviewed in full, of which 86 articles were excluded based on the following reasons: no PM2.5 data (*n* = 19), no PM2.5-increment data (*n* = 3), insufficient incidence and outcome data on liver disease (*n* = 62), and duplicate data (*n* = 2). The remaining 16 studies, which included one cross-sectional study [22], one retrospective cohort study [29] and 14 prospective cohort studies [24,30,31,32,33,34,35,36,37,38,39,40,41,42], were included in the present meta-analysis.

### 3.2. Characteristics of Studies Included in the Meta-Analysis

The characteristics of 16 studies included in the meta-analysis were shown in Table 1. All studies were published between 2016 and 2022, which included more than 331,114 participants. In the above studies which reported age, the age range of the participants was more than 18 years. Regarding the types of liver diseases, 13 studies were related to liver cancer [29,30,31,32,34,35,36,37,38,40,41,42], one study was related to liver cirrhosis [33], and two studies were related to fatty liver disease (metabolic dysfunction-associated fatty liver disease and NAFLD, respectively) [23,24]. Among them, eight studies investigated the correlationship between the incidence of chronic liver disease and PM2.5 exposure [23,24,30,31,32,33,34,35], and other eight studies observed the mortality of chronic liver disease [29,36,37,38,39,40,41,42].

Years Enrolled, data collection time; NOS, Newcastle–Ottawa Scale.Studies were conducted in multiple countries, mainly in Asia (*n* = 7), Europe (*n* = 4), and the Americas (*n* = 5). The number of participants in these studies ranged from less than 300 to more than 180 thousand. All studies included both male and female participants. The NOS score for methodological quality of the included studies ranged from 8 to 9 points, with an average of 8.9 points, and 16 studies were of high quality (NOS score ≥ 8). 

### 3.3. Overall Meta-Estimates and Publication Bias 

To explore the association between PM2.5 exposure and chronic liver diseases, the adjusted HRs and 95% CIs, which were extracted from included researches, were used to calculate the overall effects. The results were I^2^ = 73.9%, *p* < 0.001 under the random effects model. The results showed that a 10 μg/m^3^ increase in PM2.5 concentration was significantly correlated with chronic liver diseases, and HR was 1.27 (95% CI: 1.19–1.35, *p* < 0.001), indicating that maternal exposure to PM2.5 was positively correlated with chronic liver diseases. The result of the Egger’s test for asymmetry showed *p* = 0.28, indicating no publication bias (shown in Figure 2). In addition, the sensitivity analysis showed that the summary results did not substantially change when excluding any single study.

### 3.4. Subgroup Analyses of Particulate Matter on Risk of Liver Diseases

Then, we presented the association between PM2.5 and risk of chronic liver diseases in subgroup analyses by incidence and mortality. Across all selected studies, in total, PM2.5 significantly increased both the incidence and mortality of chronic liver diseases (pooled HR = 1.33, 95% CI: 1.20–1.46; I^2^ = 71.5% and pooled HR = 1.21, 95% CI: 1.09–1.35; I^2^ = 78.2%, respectively) in Figure 2. The association between PM2.5 and risk of chronic liver diseases in subgroup analyses by region and types of chronic liver disease is shown in Table 2. Regarding region, PM2.5 had a significant effect on non-lung risk of chronic liver diseases in Asia, Europe, and North America (Table 2). Moreover, the meta-analyses of higher exposure to PM2.5 and risk of liver cancer, liver cirrhosis, and fatty liver disease showed significant correlation (Table 2). 

## 4. Discussion

In the present meta-analysis, 16 studies involving more than 330 thousand participants in 13 countries were included. Our study showed a significant positive association between PM2.5 exposure and risk of chronic liver diseases. Such association was also found when studies were separated into incidence and mortality of chronic liver disease. Among each type of chronic liver disease, increments in PM2.5 exposure had a harmful impact on the risk of liver cancer, liver cirrhosis, and fatty liver disease. Therefore, we believe that the data obtained from our study show a strong correlation between PM2.5 exposure and the risk of chronic liver diseases.

Overexposure to PM2.5 pollution promoted the incidence and mortality of serious multi-system diseases including the respiratory system, cardiovascular systems, and digestive system. An increasing number of studies have proved that PM2.5 enters the blood circulation and deposits in liver, brain, and other organs through gas exchange in the alveolus [43,44]. Recent studies also reported that inhalation of PM2.5 disordered gut microbiota, leading to abnormal serum metabolome and insulin resistance [45,46]. However, the role of PM2.5 in the occurrence, development and mechanism of chronic liver diseases remains unclear.

Several potential mechanisms related to our findings have been proposed. Firstly, long-term exposure to PM2.5 induced local tissue or systemic inflammatory response through stimulating inflammatory cells [47]. Zhang et al. [48] found that PM2.5 exposure significantly elevated tumor necrosis factor-α (TNF-α) and interleukin-6 (IL-6), and caused liver inflammation and damage in rats at the same time. PM2.5 inhalation induced inflammation of liver, which caused abnormal liver function, consequently promoting NAFLD [49]. Afterwards, an in vivo study showed that inhaled PM2.5 pollutants induces steatosis and portal inflammation with increasing expression of inflammatory factors such as IL-6, TNF-α, NF-κB in the liver [50]. A prospective cohort study showed that long-term PM2.5 exposure induced occurrence of liver cancer (HR, 1.28; 95% CI, 0.88–1.92, per 12.2 μg/m^3^ PM2.5 increment) through targeting the liver with persistent proinflammation [30].

Another key potential mechanism was caused by oxidative stress. An in vivo study showed that continuous exposure to PM2.5 activated hepatic stellate cells through oxidative stress and increased the risk of liver fibrosis [51]. Ding et al. [52] explored that chronic PM2.5 exposure disordered redox homeostasis, and induced hepatic steatosis in mice by increasing expression of hepatic Nrf2 and Nrf2-regulated antioxidant enzyme gene, the sterol regulatory element binding protein-1 c, and fatty acid synthase in the liver. More and more evidence showed that the development of liver cancer was associated with oxidative stress, which induced DNA damage, cancer-related gene mutations, and dysregulation [53]. In addition, a randomized controlled trial study from China, including 107 cases of liver cancer and 178 healthy controls, confirmed the significant association between oxidative stress and the risk of liver cancer [54].

Gut microbiota dysbiosis and insulin resistance might be one of possible mechanisms. Both in vivo and prospective studies have indicated that chronic exposure to PM2.5 caused dysbiosis of the gut microbiota and may subsequently contribute to the development of abnormal glucose metabolism and insulin resistance [55,56,57,58]. Previous studies presented that gut microbiota affected bioprocessing of bile acids and production of pro-inflammatory intermediate metabolites resulting in hepatic disorders [59,60]. Another in vivo study showed that 12 weeks of PM2.5 exposure activated the c-Jun *n*-terminal kinase signaling pathway, increased insulin receptor substrate-1 phosphorylation, and caused significant liver damage with hepatic insulin resistance [61]. Moreover, insulin resistance was strongly associated with fat accumulation in the liver [62]. Thus, gut microbiota dysbiosis and insulin resistance caused by exposure to PM2.5 may result in chronic liver diseases.

Studies have reported that exposure to some chemical compounds, carcinogenic compounds, and metals also increased the risk of chronic liver disease. Polychlorinated biphenyls, as environmental endocrine and metabolism disrupting chemicals, were associated with the genesis and progression of steatohepatitis and liver cancer induced by insulin resistance, dyslipidemia, proinflammatory cytokines [63,64]. Both childhood and adulthood passive smoking contributed to higher risk of fatty liver [65]. Exposure to even low doses of bisphenol A (BPA) may adversely affect liver function of children in their later life [66]. Environmental exposure to cadmium was associated with the risk of suspected NAFLD [67]. To increase the accuracy and validity of the study, more studies are urgently needed to analyze the association between different environmental pollutants and specific chronic liver diseases.

The main strengths of our present meta-analysis are the inclusion of various of chronic liver diseases, separation incidence from mortality of chronic liver diseases, and the coverage of more participants and countries than previous studies. Previous meta-analysis focused on the relationship between PM2.5 exposure and liver cancer. However, in the present meta-analysis, we included the studies to comprehensively analyze the relationship between PM2.5 exposure and other chronic liver diseases as well. All included data were adjusted for multiple hypothesized confounders in the present study, which was also free from publication bias. Overall, as far as we know, this is the first study that has comprehensively demonstrated the relationship between different types of chronic liver diseases and PM2.5 exposure.

Additionally, our study also has several limitations. First, each included study has different adjusted confounders. Thus, there may be information bias in our study, which reduced the accuracy. Second, there was a lack of data on other types of chronic liver disease except liver cancer. Third, we did not distinguish between indoor and outdoor PM2.5 pollution, and did not explore the source and composition of PM2.5 pollution. Fourth, heterogeneity might be generated from difference between individuals and between various observational studies to influence the result. Finally, the number of included studies is not enough; therefore, more studies are needed.

## 5. Conclusions

Our results suggested that PM2.5 exposure was associated with an increased risk of different chronic liver diseases. Although data for specific liver diseases were restricted, our results suggested that PM2.5 can increase the risk of fatty liver disease, liver cirrhosis, and liver cancer. The current meta-analysis indicated that diverse biological mechanisms might play a role in the numerous subtypes of chronic liver disease. More and more studies focus on the effects of various ambient air pollution on the risk of chronic liver diseases. Therefore, future research is needed to strengthen the association between certain types of air pollutants and specific chronic liver diseases.

## Figures and Tables

**Figure 1 ijerph-19-10305-f001:**
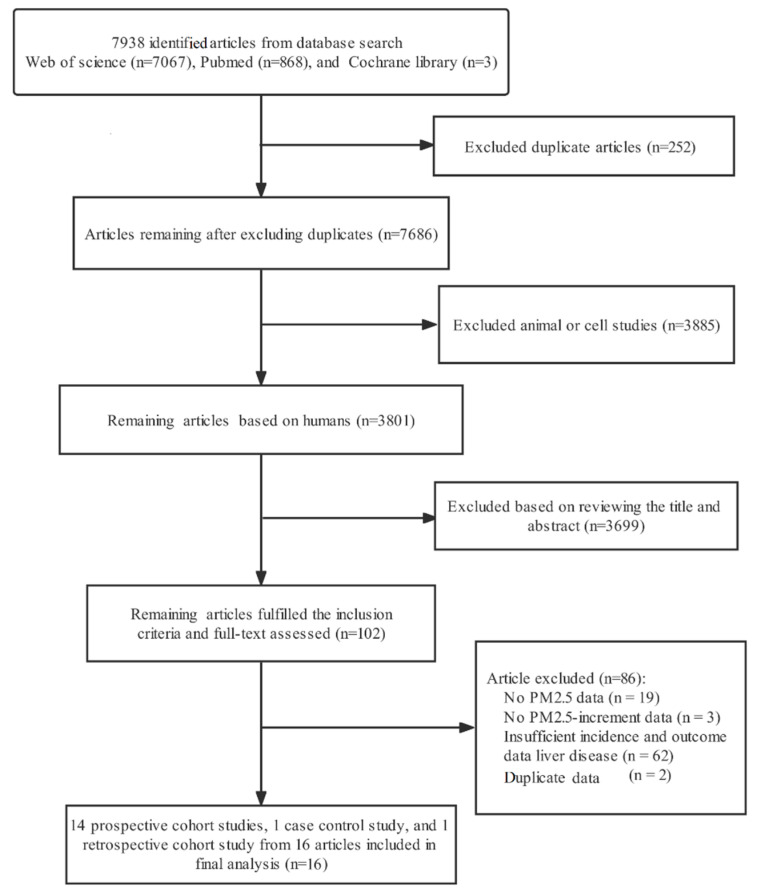
Flow diagram for identification of relevant studies.

**Figure 2 ijerph-19-10305-f002:**
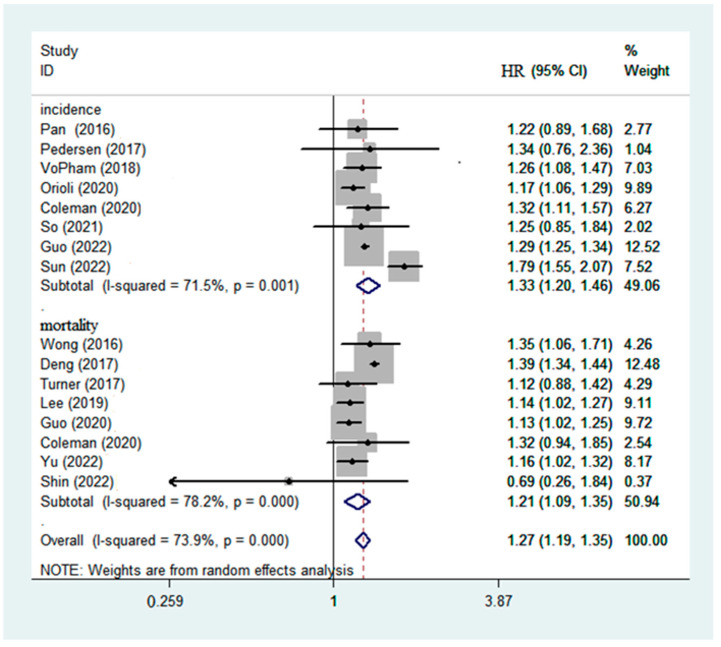
Long-term exposure to fine particulate matter (PM2.5) and risk of chronic liver disease according to in a random-effects meta-analysis. HR, hazard risk; CI, confidence interval (HR and 95% CI are for a 10 μg/m^3^ increase in PM2.5) [23,24,29,30,31,32,33,34,35,36,37,38,39,40,41,42].

**Table 1 ijerph-19-10305-t001:** General characteristics of included studies.

Studies	Study Design	Location	Years Enrolled	Age Range (Years)	Gender	Sample Size	Health Effects	Adjustment Variables	NOS
Pan et al. [30] (2016)	Prospective Cohort Study	Taiwan	1991–1992	30–65	Male/female	464	Increased incidence of liver cancer	Age 40 to 49 years, males, positive for HBsAg serostatus, positive for anti-HCV serostatus, and had alcohol consumption habit	9
Pedersen et al. [31] (2017)	Prospective Cohort Study	Denmark, Austria and Italy	1985–2005	42–57	Male/female	279	Increased incidence of liver cancer	Age (time scale), sex, calendar time smoking status, alcohol, occupational exposure, employment status, education, area-level SES	9
VoPham et al. [32] (2018)	Prospective Cohort Study	USA	2000–2014	50–74	Male/female	56,245	Increased incidence of liver cancer	Age at diagnosis, sex, race, year of diagnosis, SEER registry, prevalence of heavy alcohol consumption, smoking, obesity, diabetes; population density; median household income; percentage with a bachelor’s degree or higher; percentage unemployed; percentage of individuals below the poverty level; percentage foreign born; urbanicity; and ambient UV exposure	9
Orioli et al. [33] (2020)	Prospective Cohort Study	Italy	2001–2005	≥30	Male/female	10,111	Increased incidence of liver cirrhosis	Sex, age, educational level, occupational status, marital status, place of birth, and area-level SEP	9
Coleman et al. [34] (2020)	Prospective Cohort Study	USA	1992–2016	18–84	Male/female	185,012	Increased incidence of liver cancer	percentage of the county in various age buckets; percentage male; percentage White, Black, Hispanic, and other; percentage who did not graduate high school, graduated high school, or obtained more education than high school; median income, rent, and home value; percentage below 150% poverty; percentage working class; percentage unemployed; percentage living in a rural area; percentage smokers; percentage who consume alcohol; percentage who are physically active; and percentage of individuals in a county who are obese using LOESS models with 3 df.	9
Guo et al. [23] (2022)	a Cross-Sectional Study	China	2018–2019	41–64	Male/female	17,951	Increased incidence of metabolic dysfunction-associated fatty liver disease	Age, sex, ethnicity, education attainment, annual household income, study region, alcohol consumption, smoking status, second-hand smoke, high fat intake, low fruit and vegetable intake, physical activity, and indoor air pollution	9
So et al. [35] (2021)	Prospective Cohort Study	Sweden, Denmark, Netherland, France, Austria	1985–2015	34–62	Male/female	512	Increased incidence of liver cancer	Age (time scale), sex (strata), subcohort (strata), calendar year of baseline, smoking status, employment status, and mean income at the neighborhood level in 2001	9
Sun et al. [24] (2022)	Prospective Cohort Study	Taiwan	2001–2016	46–65	Male/female	35,614	Increased incidence of non-alcoholic fatty liver disease	Age, year of enrollment, season of measurement, gender, smoking status, alcohol consumption, occupational exposure, educational attainment, vegetable intake, fruit intake, sugar drink intake, fried food intake, habitual physical activity, physical activity at work, cancer, long-term use of hyperlipidemia drugs, cardiovascular disease, and hypertension	9
Wong et al. [36] (2016)	Prospective Cohort Study	Hong Kong	1998–2011	≥65	Male/female	676	Increased mortality of liver cancer	Age (year), Gender, BMI quartiles, Smoking, Exercise (days/week), Education, Monthly expenditure (USD)	9
Deng et al. [37] (2017)	Prospective Cohort Study	USA	2000–2009	51–77	Male/female	20,221	Increased mortality of liver cancer	Age, sex, race/ethnicity, marital status, socioeconomic status, rural–urban commuting area, distance to primary interstate highway, distance to primary US and state highways, month of diagnosis, year of diagnosis and initial treatments	9
Turner et al. [38] (2017)	Prospective Cohort Study	Canada	1982–2004	Majority: 40–69	Male/female	1003	Increased mortality of liver cancer	Age, race/ethnicity, gender stratified and adjusted for baseline values of education; marital status; body mass index; body mass index squared; smoking status; cigarettes per day; cigarettes per day squared; duration of smoking; duration of smoking squared; age started smoking; passive smoking, vegetable/fruit/fiber consumption; fat consumption; beer, wine, liquor consumption; industrial exposures; occupation dirtiness index; and 1990 ecological covariates	9
Lee et al. [29] (2019)	Retrospective Cohort Study	Taiwan	2000–2009	49–74	Male/female	1003	Increased mortality of liver cancer	Child–Pugh score, macrovascular invasion	8
Guo et al. [39] (2020)	Prospective Cohort Study	Taiwan	2001–2014	≥18	Male/female	611	Increased mortality of liver cancer	Age, sex, education, BMI, cigarette smoking, alcohol drinking, physical activity, vegetable and fruit intake, occupational exposure, season and year of enrolment	9
Coleman et al. [40] (2020)	Prospective Cohort Study	USA	1987–2014.	18–84	Male/female	761	Increased mortality of liver cancer	buckets) and categorical variables for BMI, income, education, marital status, rural versus urban, region, and survey year	9
Yu et al. [41] (2022)	Prospective Cohort Study	Brazil	2010–2018	≥20	Male/female	82,297	Increased mortality of liver cancer	The result was estimated by random effect meta-analysis with no statistical adjustment, because those models were based on the same sample.	8
Shin et al. [42] (2022)	Prospective Cohort Study	Korea	2007–2015	Mean age: 46.58	Male/female	651	Increased mortality of liver cancer	Age, sex, Health insurance premium, Employment status, Cigarette smoking status, Cigarette smoking amount (pack per day), Cigarette smoking period (year), Alcohol consumption, Physical activity, Nutrition, BMI, Family history of cancer, district-level of Elderly population, completeness of high school graduates, Gross Regional Domestic Product, and Population density, Area type, Health screening participation	9

**Table 2 ijerph-19-10305-t002:** Long-term exposure to fine particulate matter and risk of chronic liver diseases in the subgroup meta-analyses.

	Total Studies			Incidence			Mortality		
	No. of Study	Pooled HR (95% CI)	I^2^ (%)	No. of Study	Pooled HR (95% CI)	I^2^(%)	No. of Study	Pooled HR (95% CI)	I^2^ (%)
**Region**									
Asia	7	1.29 (1.14–1.45)	82.3	3	1.43 (1.11–1.84)	89.6	4	1.15 (1.07–1.23)	0
Europe	4	1.17 (1.09–1.26)	0	3	1.18 (1.08–1.29)	0	1	1.16 (1.02–1.32)	NA
North America	5	1.35 (1.27–1.43)	15	2	1.29 (1.15–1.46)	0	3	1.32 (1.16–1.50)	36.9
**Type of disease**									
Liver cancer	13	1.23 (1.14–1.33)	63.4	5	1.28 (1.15–1.42)	0	8	1.21 (1.09–1.35)	78.2
Liver cirrhosis	1	1.17 (1.06–1.29)	NA	1	1.17 (1.06–1.29)	NA	0	NA	NA
Fatty liver disease	2	1.51 (1.09–2.08)	94.7	2	1.51 (1.09–2.08)	94.7	0	NA	NA

NA, not applicable; HR, hazard ratio. Regarding geographic region, PM2.5 had a significant effect on both incidence and mortality of chronic liver diseases in Asia, Europe, and North America. The meta-analysis illustrated the association between PM2.5 exposure and risk of incidence of liver cancer, liver cirrhosis, and fatty liver disease (pooled HR = 1.28, 95% CI: 1.15–1.42; I^2^ = 0, pooled HR = 1.17, 95% CI: 1.06–1.29; I^2^ = NA, and pooled HR = 1.51, 95% CI: 1.09–2.08; I^2^ = 94.7%, respectively; Table 2).

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
