# Peer review of "Long-Term Exposure to Fine Particulate Matter and the Risk of Chronic Liver Diseases: A Meta-Analysis of Observational Studies"

_ijerph, 2022, doi:10.3390/ijerph191610305_

Round 1
Reviewer 1 Report
1. Lines 31-32: First sentence of the introduction repeats the word increase in various forms. Please rephrase and also remove the word "significant".
2. Lines 37-38: Please add a reference.
3. Lines 69-70: You searched these databases for articles published between 1990 and 2022, you did not conduct the search during this time-span. Please rephrase.
4. Inclusion and exclusion criteria: You claim that you have included epidemiological observational studies and that you have excluded in vivo and in vitro studies (which is an unspecific term and might also include observational studies). Furthermore, you do not mention anything about previous meta-analyses (as exclusion criteria). Please revise this subchapter.
5. Lines 113-123: I do not find appropriate the style of writing, I do not understand why the authors have addressed these lines to a particular person/public ("you") and began the sentences with "please".
6. Line 132: What does population sharing represent?
7. Table 1: sample size column- please mention in brackets that you refer to number of patients; why is there a NA in sample size in one study and why are there multiple NAs in terms of age limits?
8. Figure 2: There is no HR in the figure, but mentioned in the legend. The legend is lacking the acronym of ES.
9. Line 212: the world "exploded" should definitely be replaced.
10. Lines 235-237: "In addition, compared with previous studies, our study increased the effect on other chronic liver diseases and PM2.5 besides liver cancer." Please rephrase
11. Could you refer in the discussion section to the previous meta-analyses conducted?
12. You have used the term "chronic liver diseases" among the result section, yet at the end of the discussion section you claim that only data regarding liver cancer was analyzed. In the conclusion chapter, again, you mention "fatty liver, liver cirrhosis, and liver cancer.". Please detail in the result section which types of liver diseases were analyzed and pay attention to be consistent among the manuscript with the pathologies studied.
13. English language requires revision.
Reviewer 2 Report
From the perspective of methodology, many important issues and definitions are not clearly described in this study, which makes the final combined results of meta-analysis unreliable and the conclusions lack practical significance. The critical flaws are as follows:
1. The 16 studies included in this meta-analysis, including the cross-sectional study, retrospective cohort and prospective cohort study, were all observational studies, but the design principles of each scheme were quite different, and the combination would cause great heterogeneity. In particular, the effect size extracted from each scheme is different, so the combined results after forced conversion to RR or HR are not accurate.
2. The purpose of this study was to explore the relationship between long-term PM2.5 exposure and chronic liver disease, but neither of these two important variables was clearly defined in the inclusion and exclusion criteria. For example, how is "PM2.5 exposure" measured? Data from monitoring station or the researchers themselves? How Long does “long-term” mean? The outcome events refer to "morbidity" and "mortality", so does the combined results refer to risk of morbidity or mortality? The absence of these important contents in the meta-analysis makes the results of this study less valuable.
3. In addition, the standard of English is unacceptable. There are many areas where it is difficult to interpret the desired meaning. Therefore, I recommend that the manuscript is re-written in consultation with a native English speaker with physiological expertise. To give an idea of the type of language errors that require correction, I give the following examples below:
Page 1, line 33: “air pollution particulate matter” should be “ambient particulate matter”
Page 2, line 55: “Above meta-analysis studies” should be “In the above meta-analysis studies,”
Page 3, line 130: “102 articles were examined the full texts” should be “102 articles were reviewed in full,”
Page 3, line 132: “data” should be “data on”, and what dose “population sharing” mean?
Page 4, line 141: “range age” should be “age range”
Page 4, line 145: “and eight studies involved the mortality of liver disease [28, 35-41], of which studies of mortality from liver disease were all liver cancer.” there is grammatical errors in the clause
Page 9, line 150: “located” should be “conducted”; line 152-154 “The NOS score range of methodological quality of included studies was 6 to 9, the average score was 8.4. The number of high-quality studies (NOS score ≥ 8) was 15.” should be “The NOS score for methodological quality of the included studies ranged 6 to 9 points, with an average of 8.4 points, and 15 studies were of high quality (NOS score ≥8).”; line 156 “To explored” should be “To explore”; line 159 “the concentration of PM2.5 increased by 10 μg/m3” should be “a 10 μg/m3 increase in PM2.5 concentration”
Page 11, line 205: “Afterwards, an in vitro study showed that inhaled PM2.5 pollutants induces steatosis and portal inflammation with…” In vitro study, how to inhale? How to observe “portal” inflammation; line 212 “exploded” should be “explored”
Page 12, line 236-237: “our study increased the effect on other chronic liver diseases and PM2.5 besides liver cancer” should be “our study increased the influence of PM2.5 on other chronic liver diseases besides liver cancer”; line 242 “Thus, our study may have information bias and reduce the accuracy” should be “Thus, there may be information bias in our study, which reduced the accuracy”
Reviewer 3 Report
Journal: International Journal of Environmental Research and Public Health (ISSN 1660-4601)
Manuscript ID: jerph-1812009
Long-term exposure to fine particulate matter and the risk of chronic liver diseases: a meta-analysis of observational studies
Jing Sui , Hui Xia, Qun Zhao, Guiju Sun, Yinyin Cai
Abstract: Although fine particulate matter (PM2.5) is a known carcinogen, evidence of the association between PM2.5 and chronic liver disease is controversial. In the present meta-analysis study, we reviewed epidemiologic studies to strengthen the evidence for the association between PM2.5 and chronic liver disease. We searched three online databases from 1990 up to 2022. The random-effect model was applied for detection of overall risk estimates. Sixteen eligible studies, including one cross-sectional study, one retrospective cohort study, and 14 prospective cohort studies, fulfilled inclusion criteria with more than 330 thousand participants from 13 countries. Overall risk estimates of chronic liver disease for 10μg/m3 increase in PM2.5 was 1.27 (95% confidence interval (CI): 1.20–1.35, P<0.001). We further analyzed the relationship between PM2.5 exposure and different chronic liver diseases. The results showed that increments in PM2.5 exposure significantly increased the risk of liver cancer, liver cirrhosis, and fatty liver disease (hazard ratio (HR)=1.23, 95% CI: 1.14–1.33; HR=1.17, 95% CI: 1.06–1.29; HR=1.51, 95% CI: 1.09–2.08, respectively). Our meta-analysis indicated long-term exposure to PM2.5 was associated with increased risk of chronic liver disease. Moreover, future researches should be focused on investigating subtypes of chronic liver diseases and specific components of PM2.5.
Comments:
It is a topic of interest to the researchers in the related area but the paper needs minor improvements before acceptance for publication. My detailed comments are as follows:
The materials and methods in the paper works very well (Data Sources and Searches, Study selection and eligibility, Data Extraction, Statistical analyses), especially the part that correspond to Literature Quality Assessment. I would like to comment that Figure 1. (Flow diagram for identification of relevant studies) seems extremely important to me in the construction of the publication. My impression is that it should have more size in the characters and perhaps build it with another style that is easier for readers to follow.
As a suggestion, it would seem appropriate to use some additional literature in the discussion to support the importance of the meta-analysis that appears to indicate that long-term exposure to PM2.5 is associated with an increased risk of chronic liver disease. I am referring to specific bibliography that highlights the importance of certain chemical substances in the induction of liver diseases. Furthermore, as the authors of the paper say, future research should focus on investigating subtypes of chronic liver disease and specific components of PM2.5. Liver disease is a multifactorial event. The different degrees that occur in the evolution of liver disease (some of them are silent…..), can be accelerated by exposure to chemical agents, or pollution. I consider that these references would add a more realistic scenario (chemical compounds, carcinogenic compounds, work, geographic area...) to the discussion and interpretation of the results obtained.
I have not detected any excess of citations of the authors in the manuscript.
I consider, once again as I have mentioned previously, that a few minimal interventions and contributions are necessary to increase the quality (already good) of the work.
Round 2
Reviewer 1 Report
1. You still have not replaced all the NAs in table 1 or at least clarified why they are present.
2. Please rephrase the paragraph in the material and method section, claiming that you searched the literature for data published between 1990 and 2002.
3. Lines 132-133: "and two studies were with fatty liver disease." What kind of fatty liver disease? NAFLD?
4. Lines 239-241: "In addition, compared with previous studies, our study increased the influence of PM2.5 on other chronic liver diseases besides liver cancer." Your study did not increase the influence of PM2.5, as you have conducted a meta-analysis which assessed the role of PM2.5 on other chronic liver diseases besides liver cancer.
5. English language still requires revision, especially in the newly added structures and paragraphs. Particularly, there are some non-academic constructions resent throughout the entire manuscript.
Reviewer 2 Report
The authors did not provide targeted answers or modifications to question 1 and question 2, which are very important factors affecting the quality of the research.
For example, subgroup analysis can be performed on different designs. More seriously, although the study collected data on morbidity and mortality, the combined results did not specify whether it was morbidity or mortality. Therefore, it is not clear what risks HR and RR refer to.
Without methodological improvements, the results will be chaotic and worthless.
